# Determination of Vibroacoustic Parameters of Polyurethane Mats for Residential Building Purposes

**DOI:** 10.3390/polym14020314

**Published:** 2022-01-13

**Authors:** Krzysztof Nering, Alicja Kowalska-Koczwara

**Affiliations:** Faculty of Civil Engineering, Cracow University of Technology, 31-155 Cracow, Poland; akowalska@pk.edu.pl

**Keywords:** polyurethane, acoustic comfort, vibrational comfort, material properties, damping, dynamic stiffness

## Abstract

This paper is aimed at investigating the use of polyurethane mats, usually used as ballast mats, for residential building purposes. Ballast mats have features that may improve the vibroacoustic comfort in residential rooms. Their strength is certainly an advantage, along with vibration and acoustic insulation. However, the problem that an engineer has to deal with, for example in modeling these types of mats, is a limited knowledge of the material’s vibroacoustic parameters. Knowledge of these may be useful for residential buildings. This paper presents measurements of the vibroacoustic parameters of polyurethane mats, together with a suitable methodology and some results and analysis. The two main material parameters responsible for vibroacoustic protection were measured: the dynamic stiffness, which is related to the acoustic properties of the material, and the critical damping coefficient, which is obviously responsible for damping. The measurement methodology is clearly described. A total of five polyurethane materials with different densities were tested. It was possible to identify a relationship between the material density and the vibroacoustic parameters, which could offer an indication of which material to use, depending on the stimulus affecting a human in a given location.

## 1. Introduction

We live more and more in urbanized spaces where the emphasis is on quick movement from one place to another. Whether we like it or not, roads and railways must therefore be situated close to our living spaces. The proximity to infrastructure has its advantages and disadvantages. On the one hand is the proximity of work, shops or cultural centers, and on the other hand, noise and vibrations disturb our rest after a day’s work. Long-term exposure to noise and vibrations can not only be a nuisance but may also contribute to the deterioration of our health. Exposure to long-term or excessive noise can cause a range of health problems ranging from stress [1], poor concentration [2] and loss of productivity in the workplace [3,4] and communication difficulties and fatigue from lack of sleep [5], to more serious issues such as cardiovascular disease, cognitive impairment, tinnitus and hearing loss. The cardiovascular effects of long-term noise include an increase in blood pressure and heart rate [6,7]. Noise also has a negative effect on attention, working memory and episodic recall [8]. One of the worst conditions, of course, is hearing loss [9,10]. However, it is worth remembering that this happens very rarely and mainly applies to employees exposed to prolonged noise without the use of appropriate health and safety measures.

Most researchers in the context of transport impacts focus on noise as a factor that can be an annoyance and neglect the impact of vibrations, and especially the combined effect of vibrations and noise. Furthermore, low-frequency vibrations like transport vibrations are in the most dangerous range for our health. Low-frequency vibrations are vibrations in the 5–25 Hz range. They are dangerous because this frequency range is similar to the resonance frequency of the internal organs of our body [11,12]. Therefore, for vibrations in this range of frequencies, disturbing symptoms may appear, resulting from long-term exposure (see Table 1).

The combined effect of vibrations and noise may be more serious than when these stimuli are considered separately [13,14].

Therefore, one of the problems in our cities is pollution caused by noise and vibration. We want to live close to all the amenities that the city gives us, but we also want to live comfortably, without excessive noise or vibrations. We currently know three methods of reducing vibration and noise in our homes. The choice of method depends on the stage of construction of the vibration source and/or the building. When the source of vibrations (road, tram or railway) is under construction, the simplest and most common legal obligation [15,16] is to isolate it from the environment, for example by using sub-ballast mats. When the source already exists and the building is under construction, the building should be designed to meet all comfort requirements by using either vibro-insulating barriers in the ground [17] or a vibroacoustic floor [18,19]. The first of these two solutions is quite expensive, and the building owner must have the right to the land in the direction of propagation in order to build such a large structure. The second is much cheaper and requires only good selection of vibro-insulating materials. The problem lies in what material to choose and how to properly design the floor. Parameters such as the dynamic stiffness [20], related to protection against impact noise, or the critical damping ratio [21], related to protection against vibrations, may be helpful in selecting the appropriate vibroacoustic materials.

Mats used in railway or tramway construction as under-sleeper pads (USPs) are mostly tested under heavy load, due to their applications. Mainly, the three basic parameters are tested: static (C_stat_) and dynamic (C_dyn_) bedding moduli [22] and the loss tangent (tan *φ*) [23]. Static and dynamic bedding moduli are used for defining the dynamic stiffening ratio (stiffening coefficient) [24]. For residential building purposes, much better parameters to use are the dynamic stiffness ratio per unit area, which does not require the heavy loading needed for C_stat_ and C_dyn_ determination, and the critical damping ratio, instead of the loss tangent.

Mats used as USPs are tested according to their purpose. Usually, their purpose is to reduce the noise transmitted from the rails and the wheels of the railway wagon body to the environment, and which constitutes not only pollution but also annoyance. Therefore, the manufacturers of track mats provide noise-related parameters such as dynamic stiffness or impact sound pressure level. However, manufacturers also provide static and dynamic parameters related to the possible range of applications, such as the static modulus of elasticity, the dynamic modulus of elasticity, the coefficient of friction, etc. Very rarely, however, do manufacturers provide a parameter related to vibration damping, which is of key importance not only in reducing vibrations but also, through this reduction, reducing noise emissions. Getzner is practically the only manufacturer of track mats that makes data related to vibration damping publicly available. In its data sheets, the firm gives a parameter called the mechanical loss factor, which is, in fact, the loss tangent reduced to one value (not given over the whole frequency range). For example, the Sylomer SR 11 12.5 mm pad has a declared mechanical loss factor value of 0.25, while the Sylomer SR 110 12.5 mm pad has a mechanical loss factor of 0.14.

## 2. Methodology of Determining Vibroacoustic Parameters

Dynamic stiffness per unit area in the system can be obtained using Equation (1) acc. [25]:(1)$$s^{\prime }=\frac{\frac{F}{S}}{\Delta d},$$
where *S* is the area of the test sample (m^2^), *F* is the dynamic force acting perpendicularly on the test sample (N) and Δ*d* is the resultant dynamic change in the thickness of the elastic material (m).

The determination of the dynamic stiffness per unit area *s*’ of the test sample was performed by the resonance method: the resonance frequency f_i_ of the basic vertical vibrations of the mass–spring–damper system is measured, where the test sample is the spring–damper of the tested system and the mass is the mass of the pressure plate.

The critical damping ratio (D) was obtained at the post-processing stage. Based on the acceleration frequency response spectrum, the logarithmic damping decrement (*δ_i_*) was determined by the half-power method using Equation (2) [26,27] and the methodology presented in Figure 1.
(2)$$\frac{\Delta f_{i}}{f_{i}}=\frac{\delta_{i}}{\pi \sqrt{1-{\left(\frac{\delta_{i}}{2\pi }\right)}^{2}}},$$
where *f_i_* is the resonance frequency (Hz), Δ*f_i_* is the frequency range in which the maximum value of the vibration acceleration at the resonant frequency (a_R_) is equal 0.707 (see Figure 1) and *δ_i_* is the logarithmic damping decrement (–).
(3)$$\delta =\frac{2\pi D}{\sqrt{1-D^{2}}},$$

When C_stat_ and C_dyn_ are measured, a simple equation allows the dynamic stiffening ratio to be calculated [24]:(4)$$\kappa \left(f\right)=\frac{C_{dyn}\left(f\right)}{C_{stat}},$$

There is also a Equation which describes the relationship between the loss tangent (tan *φ*) and the logarithmic damping decrement (*δ*) [28,29]:(5)$$\tan\varphi =\left(\frac{\delta }{\pi }\right)\left[1-\left(\frac{\delta }{2\pi }+\ldots \right)\right],$$
(6)$$\tan\varphi =\frac{\delta }{\pi },\;for\;\delta <2$$

Equation (5) can be simplified to the form of Equation (6), which applies to small values of the mechanical loss, which is equivalent to small values of damping. When considering damping in construction, we are generally dealing with very small values.

## 3. Measurement Methodology

A single-degree-of-freedom (SDOF) dynamic system is shown in Figure 2 acc. [30]), consisting of a mass–spring–damper system.

To recreate the schema presented in Table 2 in a real-world situation, the machine shown in Figure 3 was used to perform tests to obtain the vibroacoustic parameters. This machine is primarily used for testing dynamic stiffness [25]. It consists of a dynamic exciter which applies a harmonic force through a force sensor to the load plate, with a pre-loaded static force of 0.1–0.4 N. The load plate (8 kg) was placed over the tested sample using plaster of Paris in order to reduce any unevenness of the sample. The response of the system was measured using an IEPE (integrated electronics piezo-electric) accelerometer (mechanical sensors typically made of silicon, coupled with microelectronic circuits to measure the acceleration) with a sensitivity of 100 mV/g. The detailed characteristics of the devices used in the dynamic stiffness machine are presented in Table 2.

The materials chosen for the measurements were flexible rebound polyurethane foams (Type R) with varying densities from a nominal 150 kg/m^3^ to 250 kg/m^3^ (150, 160, 180, 200 and 250 kg/m^3^). The choice was dictated by the fact that in this density range, the foam is produced by the same technological process. For each nominal density, 3 samples were tested. The actual density of samples is presented in Table 3.

The dimensions of the samples for measurement were 200 mm × 200 mm × 50 mm. This is due to the fact that machine used in this test procedure was adapted to using 200 mm × 200 mm samples. The thickness can vary, in terms of the requirements of the testing machine. Considering the resonant frequency, the thickness must be high enough to provide a resonant frequency in range of 1 Hz to 100 Hz. An example of a tested sample is shown in Figure 4. It is rebound polyurethane foam with a nominal density of 160 kg/m^3^. The actual density of the sample in the picture was 156.5 kg/m^3^.

Dynamic excitation of the testing system consisted of a force generated by the exciter modulated by a sinusoidal signal. The amplitude of the applied sinusoidal force was 0.2 N ± 0.005 N and the frequency range was from 1Hz to 100 Hz. The frequency was increased every 1 s of the measurement time by 0.1 Hz. The system response was measured using an IEPE accelerometer located at the load plate. An example of the excitation force and system response at 100 Hz is presented in Figure 5.

The results of the measured accelerations and the force presented in Figure 5 are the values measured directly by the relevant sensors. Apart from the measured frequency at a given moment, the signal also contains the self-noise of the exciter and other dynamic disturbances coming from the environment, which are not completely removed by vibro-isolation of the base plate. A Butterworth low-pass filter was used before the signal was converted to its resultant amplitude to reduce possible contamination of the acceleration values used for the analysis. Possible phase shifts due to the use of the filter were ignored as only the signal amplitude is taken into account in the results.

According to Equations (2) and (3), the critical damping ratio was calculated using the methodology illustrated in Figure 1. An example of this calculation for a sample of nominal density 160 kg/m3 is shown in Figure 6. The critical damping ratio for this example was D = 7.3% (Δ*f_i_* was estimated from Figure 6 as 6.1 Hz).

The dynamic stiffness values were calculated according to Equation (1). In order to calculate the aforementioned values, the resonant frequency was approximated using the maximum value of the response spectrum. The indication of the resonant frequency is shown in Figure 7.

## 4. Measurement Results

### 4.1. Results for Rebound Polyurethane

According to the methodology described above, the two materials parameters (dynamic stiffness and critical damping ratio) were calculated for each rebound polyurethane foam sample. The results for these two parameters are listed in Table 4.

The results are also presented in the figures. In Figure 8, the dynamic stiffness results are shown and a trend line with the scattering of the results is presented.

The tested sample densities varied between 143.0 kg/m^3^ and 264.5 kg/m^3^. They fitted reasonably well to the extreme values of the nominal density of 150–250 kg/m^3^.

As can be seen from Table 4 and Figure 8, the measured dynamic stiffness was between 11 MN/m^3^ and 40 MN/m^3^. The value of R2 (0.978) is very high, and the root mean square error RMSE of 1.498 is relatively low. Therefore, the relationship between the dynamic stiffness and the density ρ for this type of material can be given as follows:(7)s′=0.2269 ρ−21.1
where s’ is the dynamic stiffness [MN/m^3^] and ρ is the density [kg/m^3^].

The critical damping ratio can be presented in the same way. In Figure 9, the results for the damping ratio are shown, with a trend line suitable for the relationship between damping and density.

The critical damping ratio values were between 6.87% and 8.54%. The value of R2 (0.9122) is high, and the root mean square error (RMSE) of 0.001527 is relatively low. The relationship between the critical damping ratio and the density of this type of material can be shown as follows:(8)D=0.0001188 ρ +0.05466,
where D is the critical damping ratio (-) and ρ is the density (kg/m^3^).

Based on the fact that two separate quantities—dynamic stiffness and critical damping ratio—can be predicted by one parameter, i.e., the density, the dependency between dynamic stiffness and critical damping ratio is analyzed below in Figure 10.

The fitting is as good as in the previous Equations. The value of R2 (0.9483) is high and the root mean square error (RMSE) of 0.001172 is low. Hence, the Equation describing the relationship between the critical damping ratio and the dynamic stiffness for rebound polyurethane foam, with densities varying from a nominal 150 kg/m^3^ to 250 kg/m^3^, can be written as follows:(9)D=0.0004968 s′ +0.06497
where D is the critical damping ratio (-) and s’ is the dynamic stiffness (MN/m^3^).

### 4.2. Control Samples

In order to evaluate the correctness of the methodology used for measurement and calculation of the dynamic stiffness and critical damping ratio, control samples of known products were used. Due to fact that manufacturers in general do not test their products for both dynamic stiffness and damping, two different materials had to be chosen as control samples. For the dynamic stiffness measurements, Ursa TEP mineral wool was tested. For the damping, the elastomer polyurethane Sylomer SR 11 from Getzner was selected as the control sample.

Considering mineral wool, it is necessary to underline the fact that the tested mineral wool has a relatively low flow resistivity (~20 kPas/m^2^). This means that to measure the dynamic stiffness, the dynamic stiffness of the enclosed gas must be added. The dynamic stiffness of the enclosed gas s’_a_ is given by the Equation:(10)$$s_{a}^{\prime }=\frac{p_{0}}{d\varepsilon },$$
where p_0_ is the atmospheric pressure during the test (1023 hPa), *d* is the sample thickness and *ε* is the porosity of the test specimen.

The results for the Ursa TEP dynamic stiffness measurements are presented in Table 5.

An example resonance curve is presented in Figure 11.

For the Sylomer damping test, it was important to recalculate the critical damping factor D’s mechanical loss factor (estimated using Equation (5b)) to a quantity given by the manufacturer. This quantity was compared with the manufacturer’s data.

As can be seen in Table 6, the results do not agree exactly with manufacturer’s data but they are within an acceptable range of error. An example frequency graph for the Sylomer sample is shown in Figure 12.

## 5. Practical Applications

The dynamic stiffness is used to evaluate the ability of a floating floor system to reduce impact sounds in dwellings. The parameter describing the damping in such a floor is responsible for the damping of vibrations that can come both from internal sources (e.g., a machine working above the ceiling) or from external sources propagating through the ground into the building, such as traffic. Examples of floor structures with elements damping both impact sounds and vibrations are shown in Figure 13 and Figure 14. Floating floors have become increasingly popular for many types of floor coverings. The term “floating floor” does not refer to a type of flooring material, but rather to a method of installation that can be used with a variety of materials, including laminates, engineered hardwood and luxury vinyl flooring. In this method, individual planks (or in some cases tiles) interlock edge-to-edge to form a single mat-like surface that simply rests on the underlayment. It is quite different from the glue-down or nail-down methods that are still used for ceramic and stone tiles, and which were once standard for nearly all flooring materials.

Based on the dynamic stiffness results, the analysis covered the possibility of reducing impact sounds depending on the density/damping of the polyurethane used and the density of the entire floating floor system. In Figure 15, the possibility of reducing the weighted impact sound pressure level of floating floor system screeds made of sand/cement or calcium sulfate with rebound polyurethane foam as a resilient layer is shown.

The same analysis was performed for the second type of floating floor (asphalt or dry), as shown in Figure 16.

In order to provide relatively smooth graphs for the results presented above, averaging of results using nominal density was performed. The density and dynamic stiffness values were linearly averaged in each group of nominal densities. After obtaining the average density and the corresponding dynamic stiffness for each nominal density, the method in Annex C of [32] was applied to obtain the reduction in impact sound level for varying thicknesses. To create the iso-surfaces presented in Figure 15 and Figure 16, spline smoothing was performed between the points obtained using the above method. The fact that decreasing the damping ratio leads to a lower reduction of impact noise may be counterintuitive. It should be underlined that the ability to reduce impact noise in terms of building acoustics depends mainly on the dynamic stiffness of the resilient layer. The damping ratio is correlated with the density (see Euqation (8)) and the dynamic stiffness (see Equation (9)). However, it should be remembered that the correlations presented in this paper are for rebound polyurethane and applying these to different materials may be misleading. The strict influence of damping on the reduction of impact noise is not the topic of this paper.

It is worth emphasizing that both floating floor systems meet the requirements for impact sound insulation, the maximum level of which in Poland is 55 dB in multifamily houses, considering high-mass floating floors and low-density polyurethanes.

## 6. Discussion and Conclusions

This article presents a measurement methodology for the vibroacoustic parameters for rebound polyurethane foam, which could be used in floating floor systems to reduce the impact sound level and to increase the damping ratio coefficient. Although floating floor systems are widely used to reduce noise, they are not used to reduce vibration. However, as shown in the article, this type of flooring system can have vibration-reducing properties, for vibrations from both from external and internal sources, due to polyurethane’s damping properties.

Rebound polyurethane foam has comparable or even better parameters of weighted reduction impact sound pressure level than the mineral wool or elasticized Styrofoam commonly used in civil engineering [33,34]. An example comparison of the impact sound reduction ranges is given in Table 7. This leads to the conclusion that rebound polyurethane is a worthy alternative to commonly used materials as a resilient layer for floating floor systems.

It is worth noting that higher densities of rebound polyurethane tend to have relatively high damping parameters. The critical damping ratio at a level of 8% is useful when a building is located in dynamic influence zones. This value of damping indicates that rebound polyurethane can be used not only as acoustic insulation but also as a material responsible for the reduction of the vibration perception of the building’s residents [35]. Of course, it must be remembered that an increase in damping increases the dynamic stiffness of polyurethane, which leads to a decrease (sometimes acceptable) in the acoustic performance of the floating floor.

This paper presents methods for the prediction of vibroacoustic parameters of specific rebound polyurethane samples based only on their density. This leads to simplification and acceleration of the design and execution process for dwellings using this type of material.

Damping, which is helpful for vibration reduction in dynamic influence zones, is inversely proportional to the noise reduction capabilities. This leads to the conclusion that during the design process, an analysis must be conducted to determine the “sweet spot” where acoustical advantages will not overcome the vibration aspect. A method to address this problem can be found in [36].

## Figures and Tables

**Figure 1 polymers-14-00314-f001:**
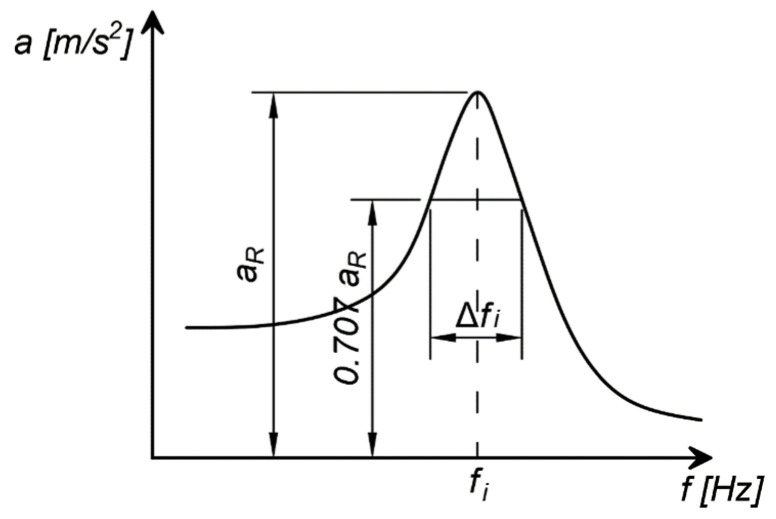
Half-power method illustration [26,27]. In order to obtain the critical damping ratio (D) an appropriate algebraic operation should be applied to the measured value of the logarithmic damping decrement (see Equation (3)) acc. [28].

**Figure 2 polymers-14-00314-f002:**
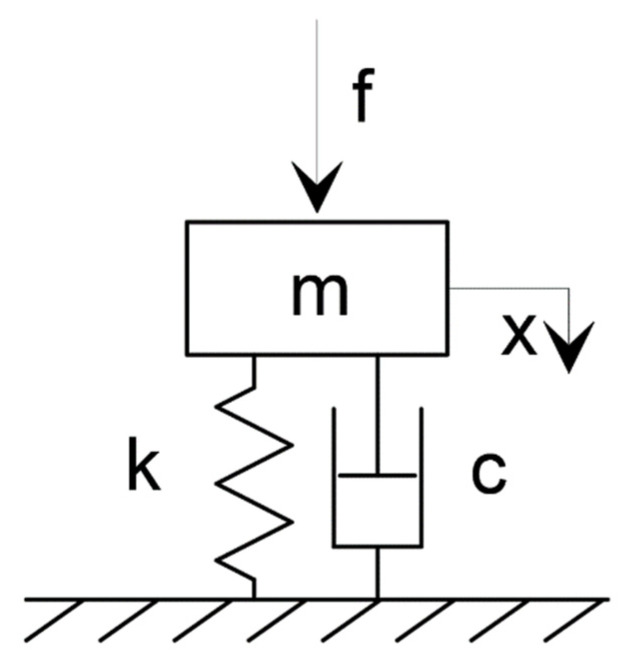
Single-degree-of-freedom mass–spring–damper system [30], where m is the mass of the system, c is the damping, k is the stiffness, f is the excitation force and x is the displacement.

**Figure 3 polymers-14-00314-f003:**
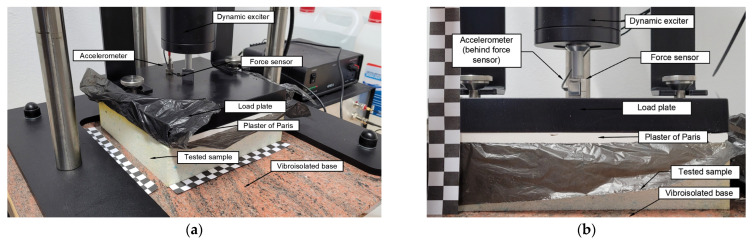
Dynamic stiffness testing machine for testing mass–spring–damper systems: (**a**) axonometry view; (**b**) front view. One tile on tile scale is 10 mm × 10 mm (own elaboration).

**Figure 4 polymers-14-00314-f004:**
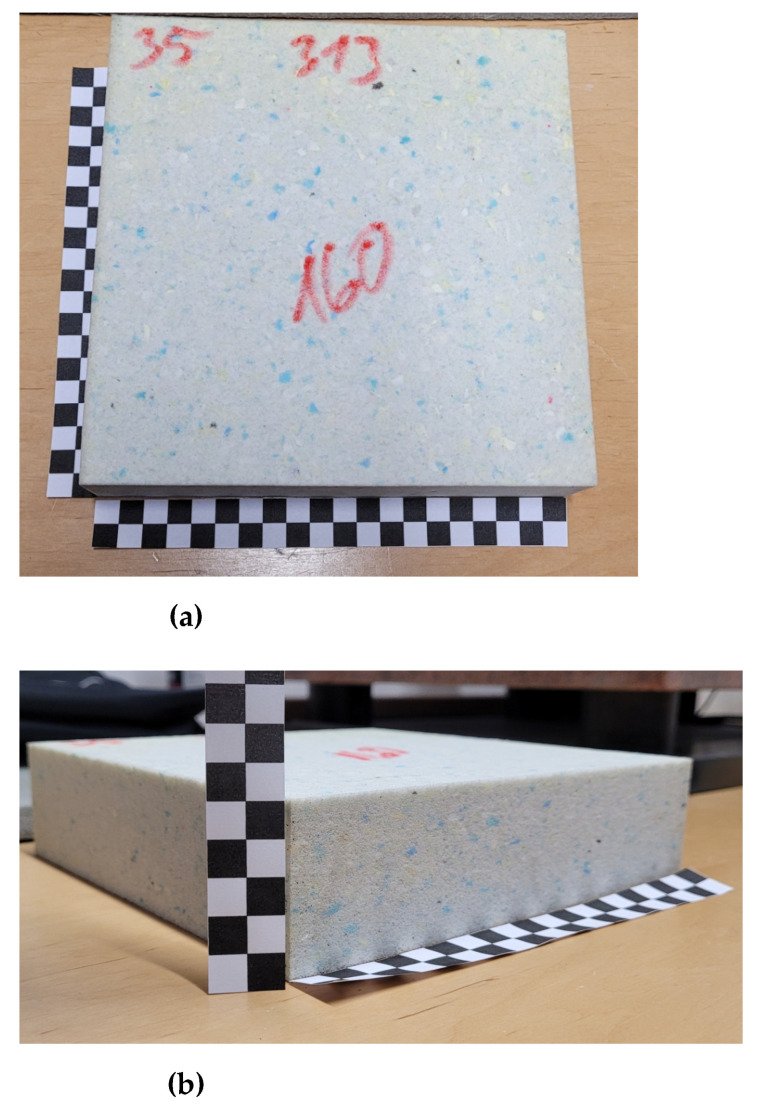
An example of tested samples of rebound polyurethane foam: (**a**) top view; (**b**) side view. One tile on the tile scale is 10 mm × 10 mm (own elaboration).

**Figure 5 polymers-14-00314-f005:**
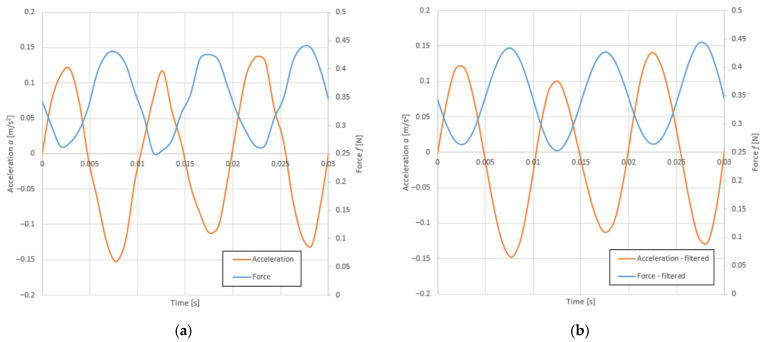
Force applied to tested system and its acceleration response at 100 Hz: (**a**) pure measured signal; (**b**) filtered signal for analysis (own elaboration).

**Figure 6 polymers-14-00314-f006:**
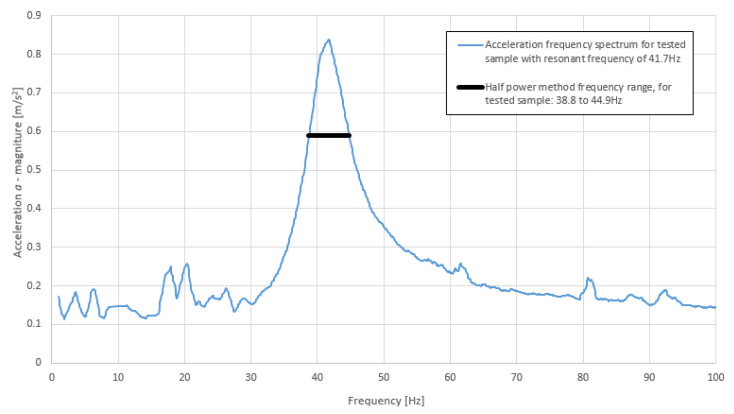
Acceleration frequency response spectrum of an example sample with indication of half-power frequency range (own elaboration).

**Figure 7 polymers-14-00314-f007:**
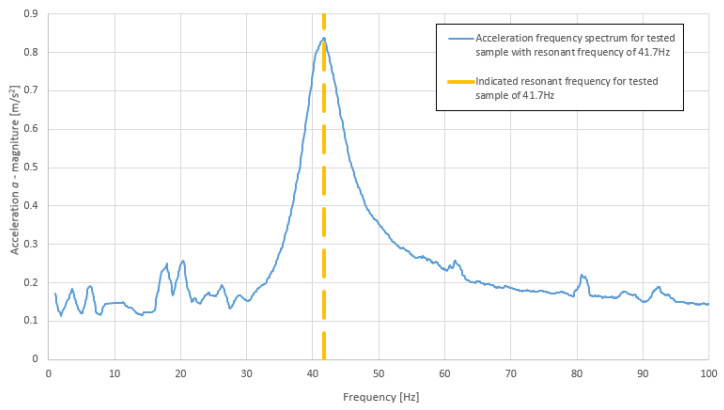
Acceleration frequency response spectrum of an example sample with indication of approximate resonant frequency (own elaboration).

**Figure 8 polymers-14-00314-f008:**
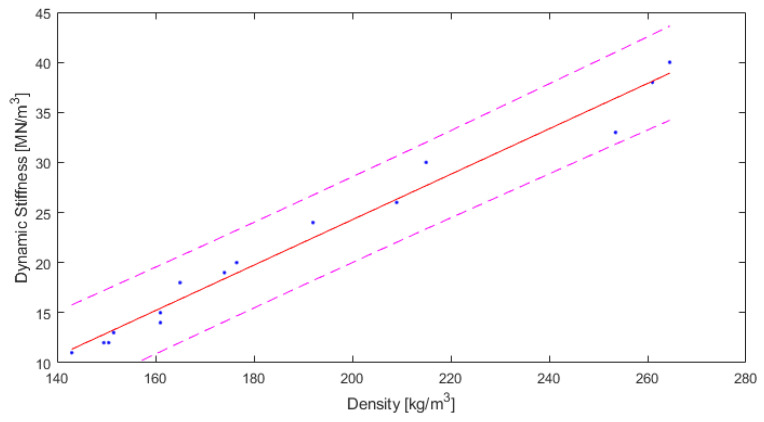
Results for dynamic stiffness depending on density of material, with fitted curve and 95% confidence bounds (own elaboration).

**Figure 9 polymers-14-00314-f009:**
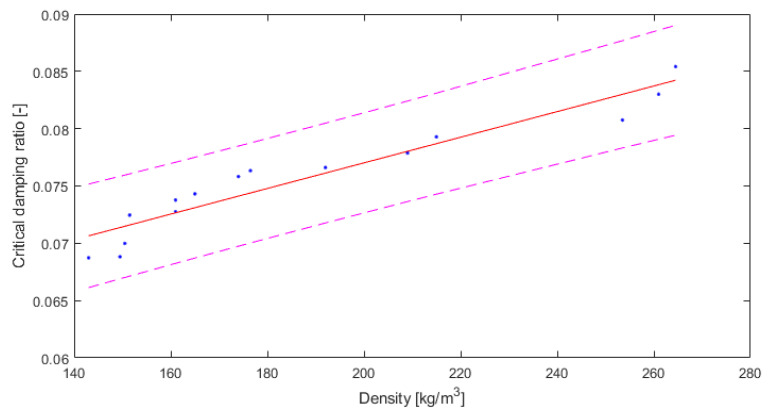
Results for critical damping ratio depending on density of material, with 95% confidence bounds (own elaboration).

**Figure 10 polymers-14-00314-f010:**
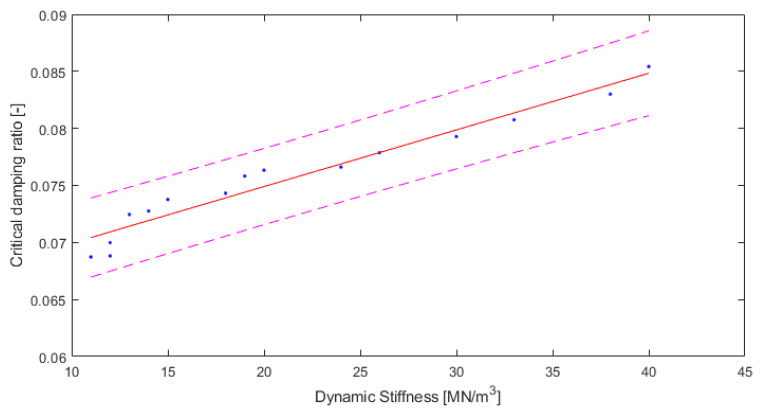
Results for critical damping ratio depending on dynamic stiffness of material, with 95% confidence bounds (own elaboration).

**Figure 11 polymers-14-00314-f011:**
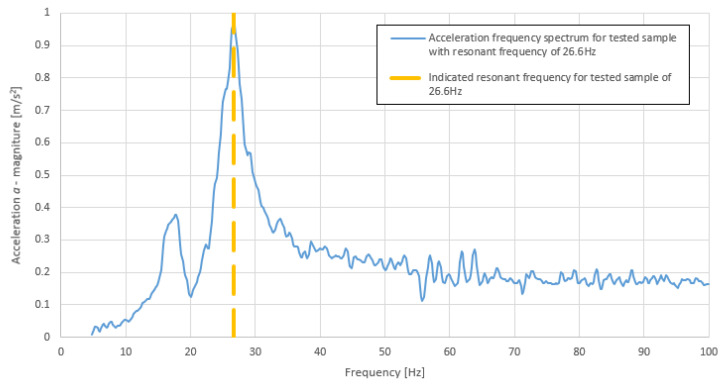
Acceleration frequency response spectrum of an example sample of Ursa TEP 23 mm mineral wool with indication of resonant frequency (own elaboration).

**Figure 12 polymers-14-00314-f012:**
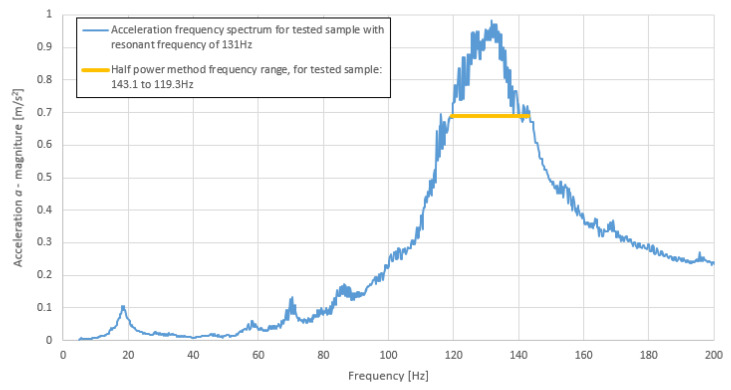
Acceleration frequency response spectrum of an example sample of Sylomer SR 11 12.5 mm polyurethane with indication of half-power frequency range (D = 9%) (own elaboration).

**Figure 13 polymers-14-00314-f013:**
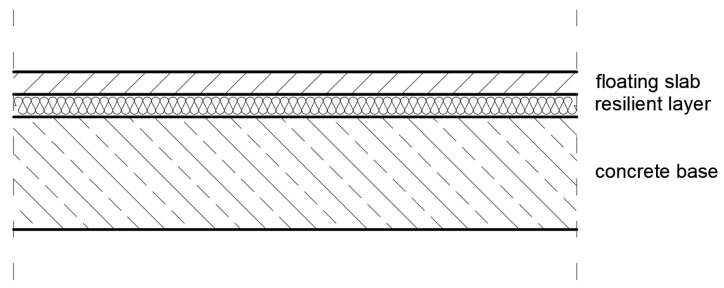
An example of floating floor system screeds made of sand/cement or calcium sulfate with rebound polyurethane foam as a resilient layer (own elaboration).

**Figure 14 polymers-14-00314-f014:**
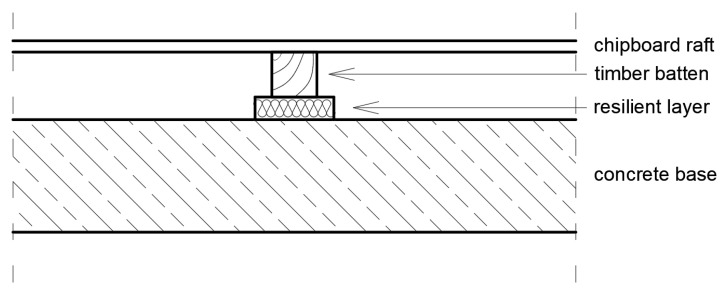
An example of asphalt floating floor or dry floating floor constructions with rebound polyurethane foam as a resilient layer (own elaboration).

**Figure 15 polymers-14-00314-f015:**
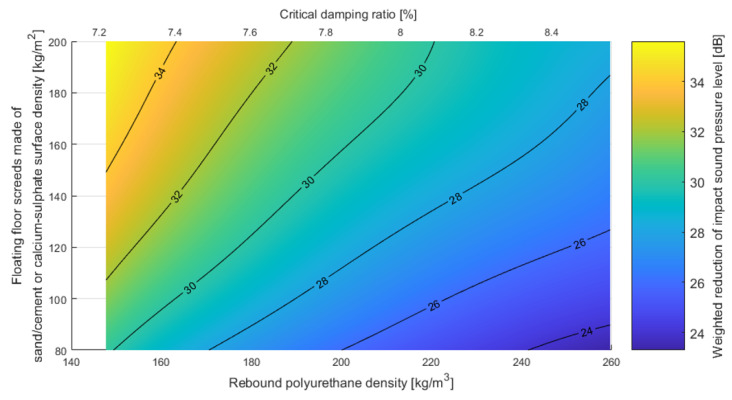
Weighted reduction of impact sound pressure level for floating floor screeds made of sand/cement or calcium sulfate with rebound polyurethane foam as a resilient layer, with varying densities and critical damping ratios (own elaboration).

**Figure 16 polymers-14-00314-f016:**
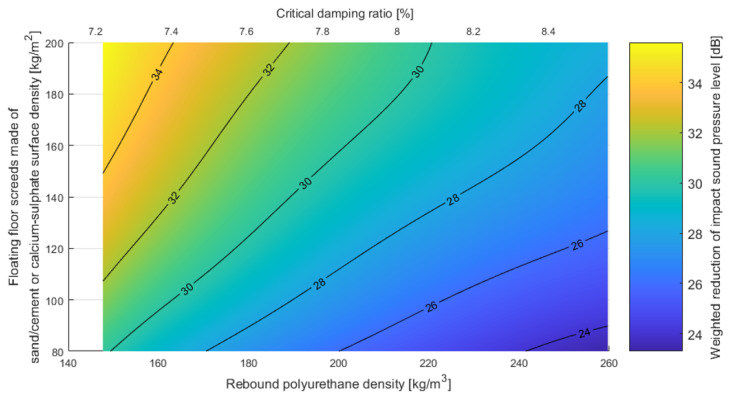
Weighted reduction of impact sound pressure level for asphalt floating floor or dry floating floor constructions with rebound polyurethane foam as a resilient layer, with varying densities and critical damping ratios (own elaboration).

**Table 1 polymers-14-00314-t001:** Symptoms caused by vibration [11].

Symptoms	*f* (Hz) ^1^
General feeling of discomfort	4–9
Head symptoms	13–20
Lower jaw symptoms	6–8
Influence on speech	13–20
“Lump in throat”	12–16
Chest pains	5–7
Abdominal pains	4–10
Urge to urinate	10–18
Increased muscle tone	13–20
Influence on breathing movement	4–8
Muscle constractions	4–9

*^1^**f*—frequency.

**Table 2 polymers-14-00314-t002:** Detailed characteristics of key components of dynamic stiffness machine.

Device Name/Manufacturer	Key Feature	Key Value of Parameters
Dynamic exciter—Brüel & Kjær Mini-shaker Type 4810	Provides sinusoidal force	Sine peak max 10 N Frequency range DC-18 kHz
IEPE accelerometer—MMF KS78B.100	Measures acceleration of system response	Peak acceleration 60 g (~600 m/s^2^) Linear frequency range (5% deviation) 0.6 Hz–14 kHz
Force sensor—Forsentek FSSM 50 N	Measures force applied to system	Capacity 50N Rated output 2.0mV/V Hysteresis ± 0.1% R.O. (rated output)
Dynamic stiffness test bench	Measures resonant frequency of sample (200 mm × 200 mm) under load of 8 kg	Linear frequency range upper limit (5% deviation) 250 Hz—measured

**Table 3 polymers-14-00314-t003:** Nominal and actual density of tested samples, with Young’s modulus (own elaboration) and Poisson’s ratio [31].

Nominal Density (kg/m^3^)	Sample ID	Actual Density (kg/m^3^)	Young’s Modulus (MPa)	Poisson’s Ratio (-) [31]
250	01	261.0	2.1	0.23
	02	264.5	2.0	
	03	253.5	1.6	
200	11	215.0	1.5	0.23
	12	192.0	1.7	
	13	209.0	1.3	
180	21	165.0	1.0	0.23
	22	176.5	0.9	
	23	174.0	0.9	
160	32	161.0	0.7	0.24
	32	161.0	0.7	
	33	151.5	0.6	
150	41	150.5	0.5	0.24
	42	149.5	0.6	
	43	143.0	0.5	

**Table 4 polymers-14-00314-t004:** Results of measurement of dynamic stiffness and critical damping ratio for tested samples of rebound polyurethane foam.

Sample ID	Actual Density (kg/m^3^)	Dynamic Stiffness (MN/m^3^)	Critical Damping Ratio (-)
01	261.0	38	0.085
02	264.5	40	0.083
03	253.5	33	0.081
11	215.0	30	0.078
12	192.0	24	0.077
13	209.0	26	0.079
21	165.0	18	0.074
22	176.5	20	0.076
23	174.0	19	0.076
31	161.0	14	0.072
32	161.0	15	0.073
33	151.5	13	0.070
41	150.5	12	0.074
42	149.5	12	0.069
43	143.0	11	0.069

**Table 5 polymers-14-00314-t005:** Results of measurement of dynamic stiffness of tested samples of Ursa TEP 23 mm mineral wool, with CI = 95% (10 samples).

Material Name	Actual Density (kg/m^3^)	Dynamic Stiffness (MN/m^3^)	Dynamic Stiffness of Enclosed Gas (MN/m^3^)	Total Dynamic Stiffness Average (MN/m^3^)	Declared Value of Dynamic Stiffness by Manufacturer (MN/m^3^)	Difference (MN/m^3^)
Ursa TEP 23 mm	81.2 (±3.1)	5.5 (±0.4)	4.6 (±0.2)	10.1	11	0.9

**Table 6 polymers-14-00314-t006:** Results of measurement of dynamic stiffness of tested samples of Sylomer SR 11 12.5 mm (17 samples).

Material Name	Actual Density (kg/m^3^)	Critical Damping Factor (-)	Mechanical Loss Factor (-)	Declared Value of Mechanical Loss Factor by Manufacturer (-)	Difference (-)
Sylomer SR 11	463.4 (±23.8)	0.096 (±0.037)	0.178 (±0.017)	0.25	0.072

**Table 7 polymers-14-00314-t007:** Comparison of impact sound pressure level reduction of rebound polyurethane with other materials. It is assumed that the floating slab is screed made of sand/cement or calcium sulfate with a surface density of 80 kg/m^2^ (according to the producers’ brochures).

Material	Range of Impact Sound Pressure Level Reduction ΔL_w_ (dB)
Tested rebound polyurethane	23–31
Mineral wool	24–34
Elasticized Styrofoam	23–29

## Data Availability

Data is contained within the article.

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
