# Peer review of "Determination of Vibroacoustic Parameters of Polyurethane Mats for Residential Building Purposes"

_polymers, 2022, doi:10.3390/polym14020314_

Round 1

Reviewer 1 Report

Dear Authors:  

I felt that the content was interesting and informative.  However, let me suggest one minor change and the two revised proposals.  

(1) Minor change: Please insert the scales into Fig.3 and Fig.4.  

(2)  The characteristics of polyurethane foams other than the densities, the production methods, (or commercialized product) should be explained more in detail.  

(3) The comparison with other potential materials used for the same purpose should be mentioned in the introduction and discussion parts.  

(4) Could you compare the results of your samples with other materials as control sample.  

Author Response

Dear Reviewer,

We would like to thank you very much for your work and your valuable comments. Thanks to this, our article will definitely become better. We hope that we have managed to dispel all doubts and correct the article as you wish it should be:

(1)Minor change: Please insert the scales into Fig.3 and Fig.4.  

The scales were added to both Figures

(2) The characteristics of polyurethane foams other than the densities, the production methods, (or commercialized product) should be explained more in detail

Table 3 was modified and additional material characteristics were added (Young modulus, Poisson’s ratio)

(3) The comparison with other potential materials used for the same purpose should be mentioned in the introduction and discussion parts.

We have added paragraphs in the introduction and discussion parts with comparison to known materials like mineral wool and Styrofoam.

(4) Could you compare the results of your samples with other materials as control sample.

We made additional measurements follow your recommendation and we revised our method on two materials: well-known mineral wool (URSA TEP 23mm) for which we have dynamic stiffness value (Table 5 and Fig.11) and with Sylomer SR11 for which we have damping parameter declared by producer (Table 6 and Fig.12)

Reviewer 2 Report

This is an interesting paper on testing the vibroacoustic performance of polyurethane mats for residential applications.

The authors should address the following issues in a revised version of their manuscript:

Line 106-107 missing references source

Line 111-112 missing source

Line 112 Figure 3

Page 4 please add the main technical data (manufacturer, accuracy, range etc) of the dynamic exciter, accelerometer, force sensor.

Figure 5: The force signal differs significantly from a sinusoidal one. Is this due to the exciter’s respose, or the measurement system? Please discuss.

Line 147 please indicate the calculation details for the critical damping ratio.

Author Response

Dear Reviewer,

We would like to thank you very much for your work and your valuable comments. Thanks to this, our article will definitely become better. We hope that we have managed to dispel all doubts and correct the article as you wish it should be:

  • Line 106-107 missing references source

The references were added

  • Line 111-112 missing source

The source was added

  • Line 112 Figure 3

The source was added

  • Page 4 please add the main technical data (manufacturer, accuracy, range etc) of the dynamic exciter, accelerometer, force sensor.

The detailed technical data was added in the text on Page 4 and in the Table 2

  • Figure 5: The force signal differs significantly from a sinusoidal one. Is this due to the exciter’s response, or the measurement system? Please discuss.

The excitation signal contains some dynamic disturbances which is explained in details in the paragraph below Fig. 5. This Figure was also improved by adding filtered signal

  • Line 147 please indicate the calculation details for the critical damping ratio.

The procedure of determining damping ratio is explained more precisely than it was in the previous version of article (lines 179-183)

Round 2

Reviewer 2 Report

can be published in the revised form